# Fabrication and Study on Magnetic-Optical Properties of Ni-Doped ZnO Nanorod Arrays

**DOI:** 10.3390/mi10090622

**Published:** 2019-09-18

**Authors:** Wei Wang, Shoulong Hui, Fuchun Zhang, Xiaoyang Wang, Shuili Zhang, Junfeng Yan, Weihu Zhang

**Affiliations:** 1School of Information Science Technology, Northwest University, Xi’an 710127, China; 17809210941@163.com; 2School of Physics and Electronic Information, Yan’an University, Yan’an 716000, China; L1463854131@163.com (S.H.); yettawang15@163.com (X.W.); zhangshuili74@163.com (S.Z.); 3Communication and Information Engineering College, Xi’an University of Science and Technology, Xi’an 710127, China

**Keywords:** magnetron sputtering, hydrothermal method, Ni^2+^-doped, ZnO nanorod arrays, magnetic-hysteresis curves

## Abstract

Zn_1-x_Ni_x_O nanorod arrays were prepared on Si substrates by magnetron sputtering and hydrothermal methods at 100 °C. We studied the effects of doped concentration and hydrothermal growth conditions on the crystal structure, morphology, photoluminescence, and magnetic properties of Zn_1-x_Ni_x_O nanorod arrays. The research results show that the Zn_1-x_Ni_x_O nanorod have the hexagonal wurtzite structure without the appearance of the second phase, and all samples have a highly preferred orientation of a (002) crystal face. The Zn_1-x_Ni_x_O nanorod arrays exhibit obvious room temperature ferromagnetism with saturation magnetization at 4.2 × 10^−4^ emu/g, the residual magnetization is 1.3 × 10^−4^ emu/g and the coercive field is 502 Oe, and also excellent luminescent properties with seven times greater luminous intensity than that of ZnO nanorod arrays. The redshift of the ultraviolet emission peak was found by Ni^2+^ doping. We further explained the source and essence of the magnetic properties of Zn_1-x_Ni_x_O nanorod arrays and deemed that the magnetic moment mainly comes from the hybrid electron exchange of O *2*p and Ni 3d state.

## 1. Introduction

Metal oxide semiconductor materials with a good optical, electrical, and chemical stability are studied deeply in application aspects of energy storage, photoelectrochemistry, and sensors [1,2,3,4]. People are devoted to the study of nano-photoelectric materials related to ZnO, and ZnO-based diluted magnetic semiconductors (DMSs) are an important branch of many research fields. Since the Dietl group [5] proposed the room temperature ferromagnetism of transition metal doped ZnO in their early theoretical work, doped ZnO has attracted widespread attention. This kind of material has been predicted to have great application prospects in spin Field Effect Transistor (FET), spin Light Emitting Diode (LED), and spin Resonant Tunneling Diode (RTD) because of their unique properties in magneto-optic and magneto-electric fields [5,6,7,8,9,10,11]. In recent decades, the application of ZnO-based diluted magnetic semiconductor materials with a wide bandgap in spintronics has attracted much attention from relevant international research fields [12,13,14,15,16,17,18]. Ni-doped ZnO is one of the most widely studied systems, and highly oriented and ordered ZnO nanomaterials are very important in spintronics and magnetic applications [19,20,21,22,23]. The applications in spintronics make a request for the ability of couple electron charges and spin degrees of freedom. Therefore, the magnetic order is particularly important at room temperature. Due to carrier-mediated exchange interaction, transition metal elements are doped in a ZnO wide bandgap semiconductor, which makes it exhibit ferromagnetic ordering at room temperature. Many other structures of 3d transition metal (TM) ions (Co^2+^, Mn^2+^, Cr^2+^, Nd^2+^) doped ZnO materials have been extensively studied [24,25,26,27,28,29,30,31], but the research on Ni-doped ZnO nanorod arrays is very limited, stimulating our research interest.

Ando et al. [32] used the MCD (magnetic circular dichroism) method to study and confirm that the 3d TM ions (Mn^2+^, Co^2+^, Ni^2+^, Cu^2+^) doped ZnO showed the function of ferromagnetism by *sp2d* electron magnetic exchange. Wei et al. [33] synthesized Zn_1-x_Ni_x_O polymorphic crystals by hydrothermal method with 0.62 wt% Ni^2+^ doped concentration and studied room-temperature ferromagnetism. Miao et al. [34] synthesized Zn_1-x_Ni_x_O columnar crystals by the hydrothermal method and indicated that the materials presented a weak paramagnetism at room temperature. Zhou et al. [35] synthesized Zn_1-x_Ni_x_O nanocombs by CVD method and concluded that the samples showed superparaferromagnetism at room temperature, which was attributed to O defects. Liu et al. [36] prepared ZnO nanorods on glass by hydrothermal method and explained that the samples showed good optical properties. Samanta et al. [37] prepared Zn_1-x_Ni_x_O nanoparticles by the chemical synthesis method, and the results proved the existence of sample magnetism and the coupling between dielectric properties and magnetoelectricity (ME). Although the experiments and theories have obtained certain achievements, the magnetic and optical properties of Zn_1-x_Ni_x_O nanomaterials still exist many disputes. Pal et al. [38] found that the absorption spectra of samples occurs a blueshift when the concentration of Zn_1-x_Ni_x_O is 3 wt%. The above research results are contrary to those of Abdel-Wahab et al. [39].

To clarify the source and essence of magnetic and optical properties of Zn_1-x_Ni_x_O nanomaterials, Zn_1-x_Ni_x_O nanorod arrays were successfully synthesized by magnetron sputtering and hydrothermal method at low temperature. The morphology, structure, elemental proportions, and optical and magnetic properties of the samples were characterized and analyzed by scan electron microscope (SEM), X-ray diffraction (XRD), energy dispersive X-ray spectroscopy (EDX), photoluminescence (PL) spectra, and magnetic-hysteresis curves (M-H curves). In addition, the electronic structure of the magnetic and optical properties of Zn_1-x_Ni_x_O nanorods arrays were systematically studied by the spin polarization density functional theory, and the magnetic coupling mechanism and magnetic source of Zn_1-x_Ni_x_O nanorods arrays were analyzed, so as to provide an experimental and theoretical basis for the experimental preparation of high quality and high temperature superconductivity (Tc) ZnO magnetic nanorods materials.

## 2. Experiment

### 2.1. Experimental Design

In our experiment, Zn_1-x_Ni_x_O nanorod arrays were fabricated by magnetron sputtering and the hydrothermal method. First, the silicon substrates were cleaned by ultrasound for 15 min with acetone, ethanol, and deionized water in turn. The first step was to prepare seed layers on monocrystalline silicon substrates by magnetron sputtering and calcine them. The second step was to mix the original solution with cobalt acetate solution of different content and mix it with magnetic stirring for 30 min to form precursor solution. The third step was to prepare vertically aligned Zn_1-x_Ni_x_O on monocrystalline silicon substrates with seed layers by hydrothermal synthesis. The precursor solution was loaded into the reactor and kept in a 100 °C temperature incubator for 3 h. After the reaction, the single crystal silicon wafer was washed three times with deionized water and alcohol in turn and then dried at 40 °C for characterization. The specific process parameters are shown in Table 1.

### 2.2. Characterization Methods

The crystal phase of the Zn_1-x_Ni_x_O nanorod arrays was investigated by an X-ray diffractometer (XRD, Bruker Advance D8, Bruker Corporation, Karlsruhe, Germany) using Cu Kα radiation operated at 40 kV and 30 mA in the diffraction angle range of 20–80° with a scanning rate of 6 deg/min and a step size of 0.02°. X-ray photoelectron spectroscopy (XPS, AXIS ULTRA, Kratos Corporation, Kyoto, Japan) characterizations were performed on a photoelectron spectrometer with a monochromatized Al Kα X-ray source. The morphologies and structures were characterized using scanning electron microscopy (SEM, Carl Zeiss SIGMA, Carle Zeiss Corporation, Jena, Germany) were performed at 3 kV. Energy dispersive X-ray spectroscopy (EDX) analysis was performed using an EDAX SDD detector (EDX, JSM-6390A, NEC Electronics Corporation, Kyoto, Japan). The photoluminescence (PL) spectra of Zn_1-x_Ni_x_O nanorod arrays were recorded by a Fluoromax-4 spectrophotometer (Horiba Jobin-Yvon, Horiba, Paris, France) in the wavelength range 325–625 nm with a 325 nm excitation wavelength. The magnetic properties were characterized and analyzed with the Quantum Design PPMS-9 (Quantum Design Corporation, San Diego, CA, America) completely liquid-free comprehensive physical properties measurement system.

## 3. Results and Discussions

### 3.1. Phase Analysis

Figure 1 shows the XRD (a) and EDX map (b) of Zn_1-x_Ni_x_O nanorod arrays. The crystallographic properties and purity of Zn_1-x_Ni_x_O nanorod arrays are clearly observed in Figure 1. The diffraction peaks of samples with different doped concentrations appear at 2θ = 34.2°, 36.5°, 47.5°, 63.7°, and 67.8° respectively, which are consistent with the hexagonal wurtzite structure of standard ZnO (No. JPCD80-0074) (P36mc, a = b = 3.2489 Å, c = 5.2049 Å). The second-phase and Ni clusters were not found in Zn_1-x_Ni_x_O nanorod arrays, which indicate that the structure of the doped samples was ZnO wurtzite with complete crystallization. The basic structure of the matrix did not change with the Ni^2+^ doping. The Zn_1-x_Ni_x_O nanorod arrays with a high purity and single phase were obtained by experiments. Especially, XRD shows that the dominant peak (002) was at the position of 2θ = 34.2° and the full width half maximum (FWHM) = 0.6. It shows that the growth of grains in the seed layer was preferential along the (002) crystal face. With the help of the Sherrer formula (D=Kγ/Bcosθ) [40], the grain size of the samples can be determined, in which K is Sheller constant (0.89), D is grain size, γ is the wavelength of X-ray Cu-Kα radiation (1.5406), B is the FWHM of each peak, and θ is the Bragg diffraction angle. Compared with our previous work [41], the (002) peak intensity of Zn_1-x_Ni_x_O nanorod arrays was larger, which indicates that the crystallinity of Zn_1-x_Ni_x_O nanorod arrays was greatly increased. With the increase of the Ni^2+^ doped concentration, the (002) peak intensity increased as a whole. The form of Ni^2+^-O^2-^ requires higher energy. However, the energy provided by the form of Zn^2+^-O^2-^ is not enough to form a Ni^2+^-O^2-^ bond. At this time, the energy changes from the surface to the interior of the crystal lattice and then generates residual stress. However, the radius of Zn^2+^ (0.74Å) [42] is larger than that of Ni^2+^ (0.69Å). The lattice parameters will inevitably decrease when Zn^2+^ replaces by Ni^2+^. Thus the intensity of the diffraction peak of Zn_1-x_Ni_x_O nanorod arrays increase further and the FWHM (B) decreases. It is concluded that the grain size (D) increases when θ is a fixed value from the Sherrer formula.

Table 2 shows the percentage of elements corresponding to the EDX spectrum of the sample (as shown in Figure 1). Among the Zn_1-x_Ni_x_O nanorod arrays, the Ni atoms content accounts for 0.14% of the total content, which is much less than the theoretical doped content. It shows that there is a loss of atoms and deposition of a small part of Ni atoms in the hydrothermal reaction process. The total content of Zn atoms and Ni atoms does not meet the stoichiometric ratio. The content of O atoms is much higher than that of Zn atoms, which indicates that there are a large number of Zn or O vacancies in the samples. This is also consistent with the results of the ferromagnetic part attributed to Zn or O vacancies.

### 3.2. XPS Analysis

The XPS spectrum was applied to confirm the element composition of Zn_1-x_Ni_x_O nanorod arrays, and explain further Ni^2+^ substitute for Zn^2+^. The survey spectra (in Figure 2a) illustrates that Zn_1-x_Ni_x_O contains Zn, O, and Ni elements. The high resolution XPS spectra for Ni, Zn, O, and C are shown in Figure 2b–e respectively. The high resolution spectrum shown in Figure 2b presents the binding energy 866.99 eV of Ni 2p states in Zn_1-x_Ni_x_O nanorod arrays. The result confirms the presence of Ni in the form of Ni 2p and does not present with the metallic form or metal oxides forms (NiO, Ni_2_O_3_, etc.), which is consistent with the XRD results. Figure 2c shows the binding energy of Zn 2p states in ZnO, which reveals 1021.79 for 2p_3/2_ and 1044.84 for 2p_1/2_ states. The peak at 529.26 eV (shown in Figure 2d) is attributed O 1s of Zn_1-x_Ni_x_O. The peak of C 1s states of Zn_1-x_Ni_x_O (shown in Figure 2e) is observed at 284.77 eV. As a result, the XPS analysis confirmed Ni in the form of Ni 2p oxidation state.

### 3.3. Morphology Analysis

Figure 3 is the SEM spectrum of Zn_1-x_Ni_x_O nanorod arrays (Figure 3a-e) and ZnO nanorod arrays (Figure 3f). It is observed that the Zn_1-x_Ni_x_O nanorod arrays are perpendicular to the substrate and exhibit a hexagonal nanorod structure, which has an obvious conical growth advantage of a positive pole (0001) and good orientation. This is consistent with the XRD results. The average diameter of Zn_1-x_Ni_x_O nanorod arrays is about 180–256 nm. Compared with the experimental results of ZnO nanorod arrays (as shown in Figure 3f) [41], it was found that the morphology uniformity of the Zn_1-x_Ni_x_O nanorod arrays was poor and the grain size was reduced. However, with the increase of Ni^2+^ doped concentration, it can be observed that the hexagonal shape of the samples tended to be more obvious and the grain size increased obviously. When the content of Ni^2+^ is 3 wt%, the morphology showed a good hexagonal shape, and the conical growth trend disappeared. Through the analysis of the growth mechanism [43,44], it can be seen that there were two different growth faces and three different growth directions in the growth process of ZnO crystal. The growth direction of C axis was much faster than that of the other two faces. The C axis was the sixth symmetric axis of the ZnO crystal. The growth speed of the other two faces was the same, which makes the top of the ZnO crystal appear as an obvious conical shape. Therefore, the content of 3 wt% Ni^2+^ might have inhibited the growth rate of the C-axis to some extent and kept the growth rates of the three crystal faces consistent. The intensity of the (002) crystal face was larger than that of the (003) crystal face and other crystal faces, which is consistent with the XRD results. In a word, the average diameter of Zn_1-x_Ni_x_O nanorod arrays increased with increasing the Ni^2+^ doped content but was significantly smaller than that of ZnO nanorod arrays, because the radius of Zn^2+^ (0.74Å) [42] was larger than that of Ni^2+^ (0.69Å). The crystal lattice parameters inevitably decreased and the grain diameter was affected when Ni^2+^ substituted for Zn^2+^.

### 3.4. Optical Performance Analysis

Figure 4 shows the photoluminescence spectra of Zn_1-x_Ni_x_O nanorod arrays. It can be seen from the spectra that the emission spectra of the samples were mainly UV emission peaks at 370 nm and a broad yellow-green defect level (DL) band ranging from 475 to 650 nm centered at 560 nm. Compared with the experimental results of ZnO nanorod arrays (as shown in sample 0 in Figure 4) [41], the luminous intensity of Zn_1-x_Ni_x_O nanorod arrays was seven times greater than that of ZnO nanorod arrays, which indicates that the formation of Ni^2+^-O^2−^ could effectively promote the luminescence of samples in the UV region. However, the intensity of the UV emission seemed to decrease as the concentration of Ni^2+^ increased. The luminescence intensity of Zn_1-x_Ni_x_O with 2 wt% Ni^2+^ doping in the UV and yellow-green region were the highest. The luminescence of samples in the UV region is generally interpreted as the recombination of free excitons (electron and hole pair) through an exciton-exciton disordered collision process, and the luminescence intensity depends on the density of free excitons [45]. The luminescence of samples in the visible region is generally interpreted as a reason of defects [46,47]. The Ni^2+^ doping can increase the defect of samples, which promote the recombination of excitons and increase the luminescence intensity. However, it requires higher energy when the Ni^2+^ enters the crystal lattice to form Ni^2+^-O^2−^ bond, so the amount of Ni^2+^-O^2−^ bond in the crystal lattice is very small, and the increase of defects can also cause the quenching of excitons [48]. As a result, the PL intensity at 550–600 nm of sample 3 was the lowest, it may be that a lot of defects of sample 3 caused the maximum quenching of excitons. The luminescence peak of the Zn_1-x_Ni_x_O nanorod arrays (shown in Figure 4) occurred as a redshift of ~2 nm compared with ZnO nanorod arrays prepared with the same process conditions (shown in Table 1), which is mainly attributed to the enhancement of the superexchange of Ni^2+^ with O^2−^ in ZnO nanorod arrays. The superexchange interaction increased the binding of exciton, which lead to a further ultraviolet luminescence peak redshift. The results studied by Yuan [49], He et al. [50], and others also show this phenomenon. We can see that the intensity of luminescence peak of Zn_1-x_Ni_x_O nanorod arrays was higher than that of ZnO nanorod arrays in the ultraviolet region and lower than that of ZnO nanorod arrays in the visible region. With the gradual increase of the Ni-doped amount, the intensity of the luminescence of samples increased first and then decreased, which indicates that the doping of Ni greatly improved the luminescence characteristics of ZnO nanorod arrays, and the mechanism of luminescence changed mainly from defect luminescence to intrinsic luminescence.

Therefore, the excellent optical properties of Zn_1-x_Ni_x_O nanorod arrays in near-band edge luminescence and defect luminescence in visible regions play an important role in explaining the magnetic properties that originate from orbital electron coupling and O vacancy, respectively.

### 3.5. Magnetic Properties Analysis

The room temperature magnetization curve (M-H curve) of Zn_1-x_Ni_x_O nanorod arrays were recorded with the change of the external magnetic field in Figure 5. It can be seen from Figure 5a that the room temperature hysteresis loop—without deducting the diamagnetism of the silicon substrate—showed a positive susceptibility when the external magnetic field was +/−5000 (Oe) and a negative susceptibility when the external magnetic field was beyond +/−5000 (Oe), which indicates that the diamagnetism at room temperature may be caused by the negative counteraction between the diamagnetism and saturation magnetization of silicon substrates. Figure 5b shows that Zn_1-x_Ni_x_O nanorod arrays had obvious room-temperature ferromagnetism behavior. The saturation magnetization was 4.2 × 10^−4^ emu/g, the residual magnetization was 1.3 × 10^−4^ emu/g, and the coercive field (Hc) was 502 Oe. Xu et al. [51] synthesized Zn_1-x_Ni_x_O nanoparticles via a low-temperature hydrothermal method, and the results show that Hc is only 139.59 Oe. Pal et al. [38] synthesized Zn_1-x_Ni_x_O (x = 0, 0.03 and 0.05) nanoparticles (NPs) by a ball milling technique and the room temperature magnetic measurements exhibit the hysteresis loop with Hc of 260 Oe. For magnetic materials used in memory/storage industry, Hc over 350 Oe under RT condition is the ideal standard [52]. Although the samples exhibited a strong coercive field, which is much larger than the standard level, the saturation magnetization was relatively weak. The results are consistent with the weak magnetic properties of Ni-doped ZnO arrays studied by Samanta et al. [37]. 

The Ni^2+^ doped ZnO nanomaterials already obtained room temperature ferromagnetism (RTFM). However, there are controversial opinions concerning the origin of RTFM, the different doped forms and the inconsistency of measurement results. Pal et al. [38] showed that ferromagnetic behavior depends on many parameters, including the concentration and distribution type of transition metal ions, defect concentration, n-type doping, p-type doping, and particle morphology. Ponnusamy et al. [53] believed that Zn vacancy (V_Zn_) and bound magnetic polaron formed by nearby Mn^2+^ interacted to form high-saturated magnetic intensity. Some people believe that the ferromagnetism behavior is attributed to the formation of some nanoscale nickel-related secondary phases, such as Ni precipitation and NiO. In our experimental results, XRD results show that the NiO phase does not appear, which mainly is attributed to the antiferromagnetism of NiO at 520 K Neel temperature [35,54]. In addition, when Ni^2+^ with the same valence as Zn^2+^ enters the ZnO crystal lattice, it cannot introduce carriers, and ZnO can only interact with other acceptor defects. As a cation, Ni^2+^ has an effect on the cation percolation threshold in ZnO nanorod arrays. When the content of Ni^2+^ increases to the threshold value, Ni atoms can come close to each other, and super-exchange interactions between Ni^2+^ ions will increase the contribution to the magnetism of the samples [55]. The ferromagnetic behavior of Zn_1-x_Ni_x_O nanorod arrays could be considered as the exchange interaction between the free delocalized carriers (hole or electron) or the localized d-spins on the Ni ions [56]. However, according to the results of the PL, M-H curves, and EDX, the content of Zn atoms is much higher than that of O atoms in the samples, which indicates that large Zn interstitials and O vacancies existed in samples. It can be inferred that the ferromagnetism of Zn_1-x_Ni_x_O nanorod arrays is mainly attributed to the bound magnetic polaron (BMP) produced by the coupling between the O atoms, defects, and the Ni^2+^ 3d orbital electron around them [57,58,59].

A strong coercive field was detected in Zn_1-x_Ni_x_O nanorod arrays, which indicates that the sample has a strong resistance to demagnetization, which can further improve the application of the material in magnetic storage media. The research results also confirm the application value of Zn_1-x_Ni_x_O nanorod arrays in permanent magnet materials, such as hard disks and disk tapes.

## 4. Conclusions 

In this paper, Zn_1-x_Ni_x_O nanorod arrays were prepared successfully by hydrothermal synthesis and magnetron sputtering method. The results show that Zn_1-x_Ni_x_O nanorod arrays with strong crystallinity kept the hexagonal wurtzite structure well. The morphology of the samples was dense and uniform, and the grain size increased obviously. The Zn_1-x_Ni_x_O nanorod arrays exhibited obvious room temperature ferromagnetism with a strong coercive field and also excellent luminescent properties with seven times greater luminous intensity than that of ZnO nanorod arrays. Therefore, Zn_1-x_Ni_x_O nanorod arrays with excellent optical and hard magnetism properties are the significant basis for further research on diluted magnetic semiconductor materials and devices applying for magnetic storage media.

## Figures and Tables

**Figure 1 micromachines-10-00622-f001:**
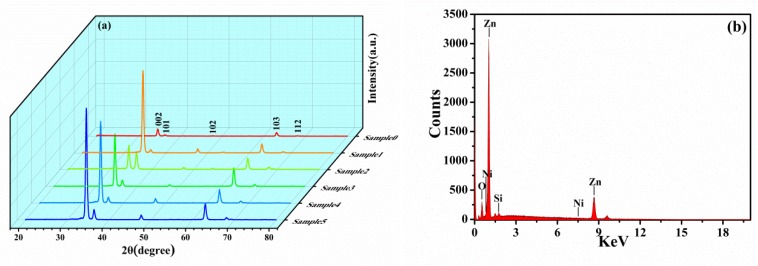
XRD (**a**) and EDX (**b**) of Zn_1-x_Ni_x_O nanorod arrays.

**Figure 2 micromachines-10-00622-f002:**
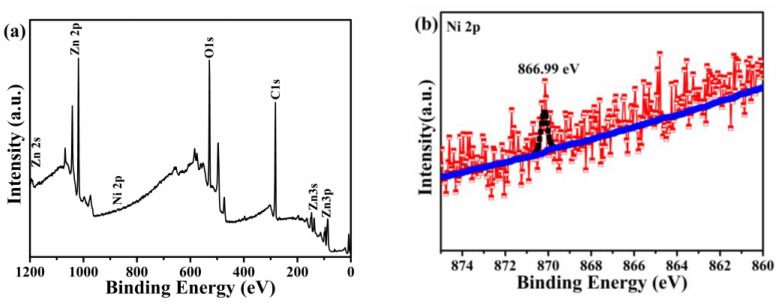
XPS of Zn_1-x_Ni_x_O nanorod arrays: (**a**) Survey spectra, (**b**) Ni 2p, (**c**) Zn 2p, (**d**) O 1s, and (**e**) C 1s core-level spectra.

**Figure 3 micromachines-10-00622-f003:**
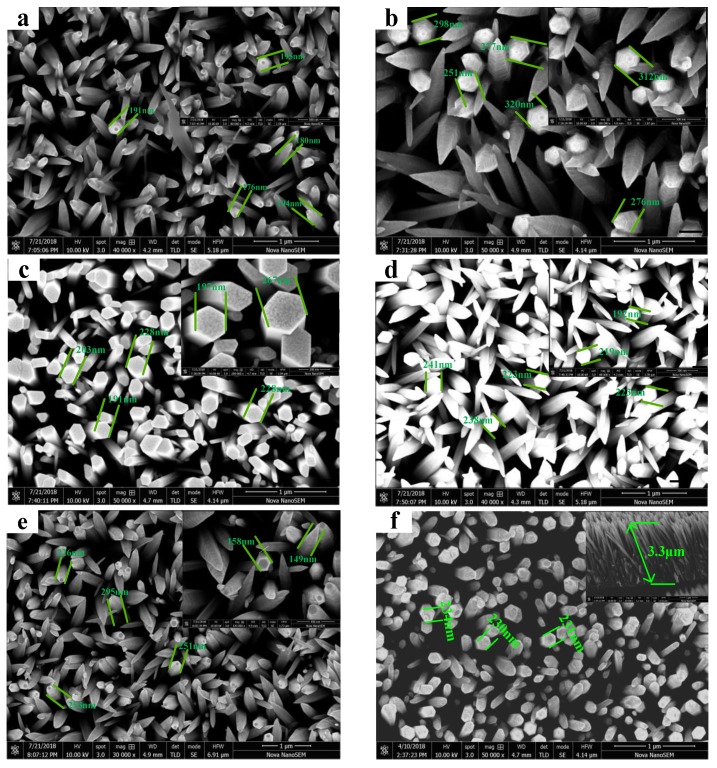
SEM of Zn_1-x_Ni_x_O nanorod arrays. (**a**) 1 wt% Ni^2+^, (**b**) 2 wt% Ni^2+^, (**c**) 3 wt% Ni^2+^, (**d**) 4 wt% Ni^2+^, (**e**) 5 wt% Ni^2+^ and (**f**) 0 wt% Ni^2+^.

**Figure 4 micromachines-10-00622-f004:**
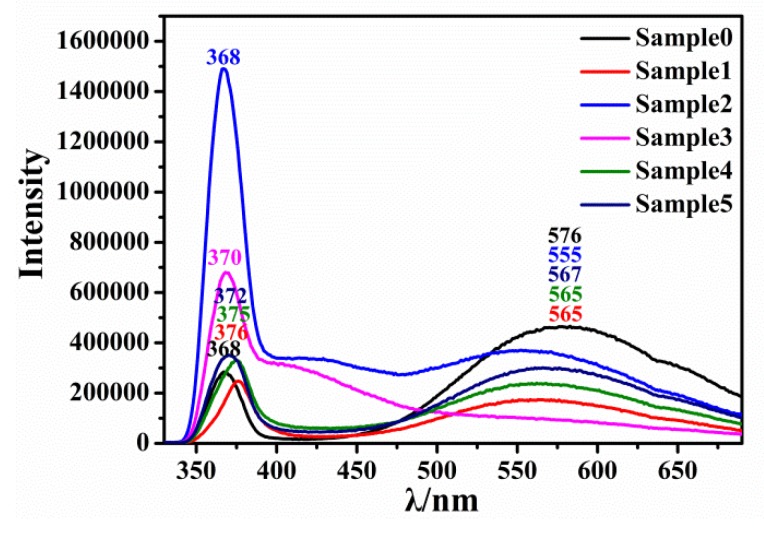
Photoluminescence (PL) of Zn_1-x_Ni_x_O nanorod arrays.

**Figure 5 micromachines-10-00622-f005:**
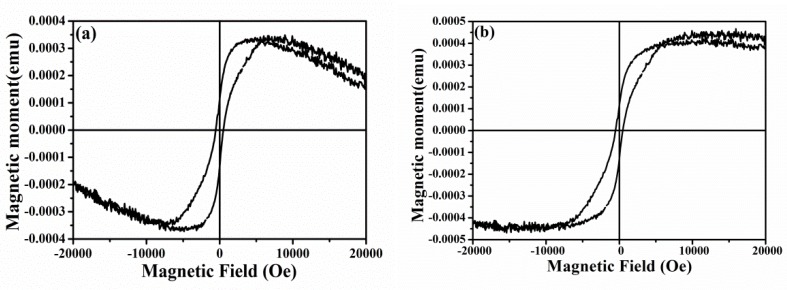
Room temperature M-H curve of Zn_1-x_Ni_x_O nanorod arrays. (**a**) Without deducting the diamagnetism of silicon substrate; (**b**) room temperature M-H curve after deducting the diamagnetism of silicon substrate.

**Table 1 micromachines-10-00622-t001:** Processing parameters of Zn_1-x_Ni_x_O nanorod arrays with different concentrations.

Factors	OH^−^/Zn^2+^	Zn^2+^ Concentration (mol/L)	Temperature (°C)	Ni^2+^ Doped Amount (%)	Time (h)
Sample 0	10	0.125	100	0	3
Sample 1	10	0.125	100	1	3
Sample 2	10	0.125	100	2	3
Sample 3	10	0.125	100	3	3
Sample 4	10	0.125	100	4	3
Sample 5	10	0.125	100	5	3

**Table 2 micromachines-10-00622-t002:** The percentage of elements of Zn_1-x_Ni_x_O nanorod arrays.

Factors	O Atom%	Zn Atom%	Ni Atom%	O/Zn Ratio
Content	20.97%	50.57%	0.14%	0.414

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
