# Peer review of "Fabrication and Study on Magnetic-Optical Properties of Ni-Doped ZnO Nanorod Arrays"

_micromachines, 2019, doi:10.3390/mi10090622_

Round 1
Reviewer 1 Report
In this work, authors showed the morphology, structure, magnetic and optical properties of Zn1-xNixO nanorods arrays by SEM, XRD, EDX, PL and magnetic-hysteresis curves. However, no further interpretation about physics and materials of Ni-doped ZnO Nanorod, and no discussion on the correlation between these analysis results. Besides, the reference sample "undoped samples" did not mention in the work for comparison. I did not recommend this work to be published.
There are some queries and comments:
Some mistyping and Grammar error. In Fig. 2, the scale of SEM is different. If add XPS to explain the Ni2+ substitute, the results should be more powerful. Page 6, line 183: redshift ~2 nm is not a comparison between these results. The comparison between the intensity of PL between samples is not good scientific analysis. Not enough information about the contribution of each peak in the PL spectra. No "undoped samples" PL spectrum to compare the doped samples. About the ferromagnetic property, there is no enough explanation and further model.Author Response
Please see the attachment.

Reviewer 2 Report
The paper is interesting, the analysis is well written and the results are supported by the data. Therefore I accept for publication the paper in its present form.
Reviewer 3 Report
This manuscript from Wang etc. reported Zn1-xNixO nanorod arrays were prepared by two step methods and exhibited strong luminescent and ferromagnetism. This paper is interesting and well-written. I would recommend the publication of this article in Micromachines, after the following comments are addressed:
The author mentioned that their Zn1-xNixO nanorod arrays exhibits strong luminescent properties compared to the ZnO nanorod arrays. They are not the same batch of samples, so the luminescent compare is not very scientific. In Figure 3, why the PL intensity at 550-600 nm of sample 3 is the lowest? In paragraph 3, page 6, the typo error “The The”. Please carefully check the whole manuscript.
Round 2
Reviewer 1 Report
Authors provided enough data to complete their results and supplied more illustration to support their deduction in the revised manuscript. In their response letter, they replied to my queries and comment, which also reflects to the revised manuscript. I can accept the present version.